# Materiality and Images: Ameghino's Collection of La Antigüedad del Hombre en el Plata in the La Plata Museum

Cecilia Simón [1,*], Mariano Bonomo [2], Sonia Lanzelotti [3,4] and Gabriel Acuña Suarez [4]

1. Departamento de Humanidades, Universidad Nacional del Sur, Bahía Blanca 8000, Argentina
2. CONICET—División Arqueología, Facultad de Ciencias Naturales y Museo, Universidad Nacional de La Plata, La Plata 1900, Argentina
3. CONICET—Instituto de las Culturas (IDECU), Facultad de Filosofía y Letras, Universidad de Buenos Aires, Ciudad Autónoma de Buenos Aires 1091, Argentina
4. SIGAPH—Grupo de Estudios Sobre Sistemas de Información Geográfica en Arqueología, Paleontología e Historia, Universidad Nacional de Luján, Luján 6700, Argentina
* Correspondence: cecilia.simon@uns.edu.ar; Tel.: +54-9-291-436-3229

**Abstract:** Florentino Ameghino's works were fundamental for archeology in Argentina. His first collections were shown in the book *La antigüedad del hombre en el Plata* {Antiquity of Man in the La Plata Basin} (1880–1881), a synthesis of the previous prehistoric studies whose purpose was arguing in favor of an ancient antiquity of the human occupation in the Argentine Pampas, which granted his archeological investigations international recognition. In this article, we present the first results of a comprehensive study on the aforementioned collection, which is currently in the Archeology Division of the La Plata Museum. We identified the pieces based on the published images, we analyzed the lithic and ceramic artifacts and bone remains, and we contextualized the information on the archeological sites from which they came. We recognized the importance of the collection for this investigation and for its conservation, acknowledging work practices related to the identification, classification and communication processes of the materials in the early days of this discipline. The revision of this museum collection shows the constructive nature of the cultural heritage, reflecting on the subjects, wisdoms, and actions that take part in their furtherance throughout time.

**Keywords:** archaeology collections; lithic, ceramic and bone materials; visual devices; museum catalogues; Florentino Ameghino





## 1. Introduction

A substantial amount of the archeological materials shown in the book *La antigüedad del hombre en el Plata . . .* (hereafter *La antigüedad . . .* ) [1,2] are part of the first collections of cultural objects obtained by Florentino Ameghino (1853–1911) during the second half of the 19th century. The study of these materials, together with the works of other researchers, such as Hermann Burmesiter (1807–1892) and Francisco P. Moreno (1852–1919), are recorded at the beginning of scientific archeology in Argentina. The impact of Ameghino's book and his work in the Argentinean archeology is remarkable. In fact, the Asociación de Arqueólogos Profesionales de la República Argentina {Argentine Association of Professional Archeologists} established Florentino Ameghino's birth date (18 September) as the Argentinean archeologist day.

The aim of this article is to present the first results of a comprehensive study on these pieces that are part of the Ameghino Collection and are currently in the Archeology Division of the La Plata Museum (La Plata, Argentina).

Between 1869 and 1877, Ameghino lived and worked as a school monitor in the city of Mercedes, Buenos Aires Province. As he explored the Rio Luján surroundings, he took his first steps towards the collection and systematization of fossils and archeological materials, pieces which he obtained, organized, and sold, as other natural scientists and enthusiasts

of the Pampas plains [3]. When he was assembling his first collections, he also read about Natural Sciences, Geology and Paleontology, advised by other scholars for the correct extraction and conservation of the bones and ancient objects, as well as for their sale and exhibition. Among these academics, we especially highlight the Italian natural scientist, Giovanni Ramorino (1841–1876), who had particular experience on the analysis of the fossil fauna with apparent marks of human action, which contributed to Ameghino's visual training [4,5]. Those first fieldworks provided materials to discuss the old antiquity of the human occupation in the Argentine territory. Ameghino was convinced that most of the bone remains of the Pleistocene fauna, mainly represented by extinct megamammals with a weight of over a ton, were clear evidence of anthropic modifications, such as the one shown in the bones with burns, striations, furrows, and fresh fractures.

Having that in mind, he presented his collections in the Exposition Universelle de Paris (1878), as well as in some of the international congresses held for such occasion, where the specialist commissions, composed of members of the National Museum of Natural History or the Society of Anthropology of Paris, such as Gabriel de Mortillet (1821–1898), Paul Broca (1824–1880), Émile Cartailhac (1845–1921) and Henri Gervais (1845–1915), accepted the materials as evidence of the coexistence between humans and the extinct megafauna [1] (p. 11). His stay in Europe allowed Ameghino to take an active part in scientific circles, discussing the evidence and raising the issue of the stratigraphy of the archeological site of Chelles (Seine-et-Marne), located in the outskirts of Paris. In turn, he created a collection of lithic materials and casts from the Lower, Middle and Upper Paleolithic, Neolithic and from the Age of Metals in Europe, which came from prehistoric sites that are now classics of archeology literature. In the case of France, it regarded the sites of Saint-Acheul and Amiens in the Somme River; Chelles and Joinville in the Parisian region; Le Pla in the Pyrenees, and Laugerie Basse, Laugerie Haute, Le Moustier, La Quina, Bergerac and Brive, in the Périgord region. As for the materials from Belgium, they belonged to the sites of Spiennes, Mesvin, Hélin, Boncelles, Strépy, Trou du Sureau à Montaigle [6,7]. The collected pieces were classified as eoliths or artifacts attributed to the Chellian, Acheulean, Mousterian, Aurignacian, Solutrean and Magdalenian "industries" [8].

Ameghino's different work stages took a graphic form through the publication of reports in conference proceedings [9,10], general and specific catalogs [11,12], and specialized magazines [13–15]. In those documents, the meetings, discussions, and conclusions were reviewed, and the illustrations of the studied archeological materials and sites were usually attached to them. This allowed him to produce new knowledge from the visual delivery of his research results, through the image design, as well as coordinate the logistics of the editing processes and obtain funds for such purposes, all of which is particularly shown in the development of his most emblematic work at the time: *La antigüedad . . .* [1,2].

Back in Buenos Aires during mid-1881, and after receiving auspicious criticisms on his investigations, Ameghino sought to become part of the local scientific circles. His main goal was to create an anthropology and archeology museum based on his collections. However, the national political context hindered those expectations, and there came to be a sale or donation of the materials to the La Plata Museum in 1886, an institution which had been inaugurated in those years. His participation as Deputy Director was a part of a temporary alliance he maintained with the founder and director of the museum, Francisco P. Moreno. As of 1888, Ameghino was no longer in the institution, although those first collections were included in the different sections of the museum [16–19]. His resignation and later dismissal stemmed from various reasons that were linked to the personality of both scholars and their interests, which were sometimes opposing, regarding the allocation of resources and the setting of priorities within the museum. From Ameghino's perspective, the issue was the delay in the delivery of the results of his fieldworks related to the discussions on the antiquity of the geological formations and human occupation [17,18,20]. During the first decade of the 20th century, these debates gained strength due to the new investigations that he was carrying out as Director of the Natural History Museum of Buenos Aires {now Bernardino Rivadavia Natural Sciences Argentine Museum} [21,22]. In this

context, the original collections in the La Plata Museum were re-analyzed by the staff of the museum and other local and international institutions, such as the Smithsonian Institution (Washington, DC, USA) [23,24].

By the 1930s, investigations about "the antiquity of men" as the hot topic in local archeology were decreasing, partly due to the alleged fraud on the controversial findings on the "Fossil Man of Miramar" in the Pampean Atlantic Coast [25–27], but also because an African origin for the hominids [28] and the introduction of anatomically modern humans to the American continent through the Bering Strait [29] were starting to be established. As of that moment, and throughout the several subsequent decades, the archeological pieces of the Ameghino Collection lost the importance they had in the previous years in the study of the origins of humankind. While those pieces were embedded on the different sections of the museum, the course of those materials was linked to the history of the institution and of the archeology, anthropology, and paleontology practices, and they were cataloged, reclassified, exhibited, or dismissed, in accordance with the demands inherent to the investigations and the building necessities.

In these last decades, there was a revision of the human remains and of the archeological materials of the Ameghino collections that are in the La Plata Museum and in the Bernardino Rivadavia Natural Sciences Argentine Museum [30–38]. We highlight an article by one of the authors (SL), who recently published a thorough analysis on the current state of part of the Ameghino collection of the La Plata Museum, recognizing several pieces published in *La antigüedad* . . . , identifying different archeological sites from which they had been collected [39]. Ameghino's collection and first works have also been analyzed from a renewed historiographic framework of the history of sciences, considering the practices as actions placed by or through objects that intervene in the creation of the collections and the institutions that are related to the study of archeology [40,41]. The results that we show on this paper stem from the analysis of the images produced by Ameghino in order to deliver the results and make his materials known and they are aligned with these new historiographic perspectives for the discipline.

## 2. Ameghino's Archeological Collection Illustrated on *La Antigüedad* . . . : Identification and Rearrangement

*La antigüedad* . . . is a synthesis published by Ameghino in Paris, between 1880 and 1881, in French and Spanish. In that work, he showed his first investigations on Argentine prehistoric archeology. A direct precedent is the catalog that he had developed for the Exposition Universelle, in 1878, in which the collection materials appeared in an arranged manner and were numbered by period and age (see below), with a temporary structure which was quite similar to the one he would resume on *La antigüedad* . . . The original edition consisted of two volumes divided into book sections and chapters, which also had 25 plates with a total of 673 figures, as well as table of plates and indexes. The published information came from various sources, including historical and linguistic studies, and travelers' and natural scientists' chronicles. It was also retrieved from archeological and paleontological materials from his collection, and from other collectors and experts, which he had access to due to his participation on the Exposition Universelle.

As for the materials that belonged to Ameghino's collection and that appear as illustrations on *La antigüedad* . . . , at present, many of them are in storage in numbers 6 and 25 of the Archeology Division and also of the Anthropology and Paleontology Divisions of the La Plata Museum. Between 2016 and 2021, we carried out the identification, rearrangement and analysis, recording a total of 191 objects that were originally described and presented as figures on Ameghino's book. For that purpose, we investigated those collections related to his work during the first years the museum was operating, and the materials were compared to the original illustrations. In some cases, we were able to identify the part of the drawn piece, as well as other parts that belonged to that same object, but that had not been illustrated. These choices, regarding the illustration and visual communication

process, provide additional information about the argumentative decisions and the quality degree or composition of the part of the represented piece.

Some of the identified pieces are labeled with different numbers, writings, and colors, which shows the activities that they were part of over time. We recognized three criteria for the acronyms, some of which were used for the same piece (Figure 1). The oldest of these numbers corresponds to the Catalogue spécial de la section anthropologique et paléontologique de la République Argentine a l'exposition Universelle of 1878 [12]. In those objects, the criterion is defined by numbers written in black ink over white labels or directly over the piece surface. We can see an example of this criterion in Figure 1, numbered as 660. In some instances, those numbers comprise grouped series depending on the material type and origin, as in the case of pottery and the bone material that are listed from number 495 to 841, or the ceramic from the Cañada de Rocha archaeological site, from number 2026 to 2644. Within this group, the characteristics and types of objects match with the brief descriptions from that French catalog.

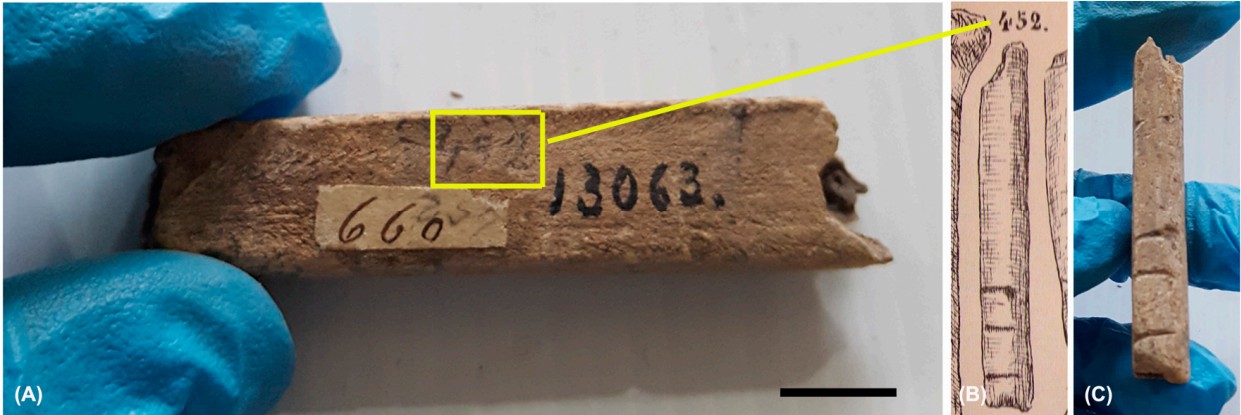

**Figure 1.** An example of the different criteria for cataloging the pieces (**A**). The number 660 is the oldest one and it corresponds to the *Catalogue spécial de la section anthropologique et paléontologique de la République Argentine a l'exposition Universelle* of 1878. The number 432, which is written in pencil and appears twice on the bone, is a number taken from *La antigüedad del hombre en el Plata* (**B**). Picture (**C**) shows the original piece from the collection. Finally, the number 13,063 corresponds to the catalog designed by Luis María Torres in 1915: *Catálogo de las colecciones arqueológicas y etnográficas*. Scale bar: 1.5 cm.

The second type of acronym corresponds to the number of the figure published on *La antigüedad* . . . This label is on the objects of plates from number XIX to XXV, written in pencil or white ink on the pieces. This criterion is represented twice in Figure 1, as depicted by number 432 on the bone and over the white label. In the case of the materials that are found on the Anthropology Division, that kind of piece shows that they have probably been stored there since 1907, a time in which Robert Lehmann-Nitsche (1872–1938) was in charge of that division and published a paper that included the description of some of those objects [23]. It is an article, with the collaboration from several academics, in which he discussed the investigations with regard to the human presence in the ancient Pampean geologic formation, describing some of the objects of Ameghino's collection. At the time of our revision, these materials, which were stored in the Anthropology Division, had little labels with the letterhead of the museum and with the following inscriptions: "Col. Fl. Ameghino. La antigüedad del hombre" and "Col. Fl. Ameghino v. Nouv. Rech. Fig. 63". In this last case, number 63 shows that which was assigned to the figure of the publication by Lehmann-Nitsche, a photograph of a set of four lithic artifacts that were preserved until now in the same box with its reference. Moreover, as for the acronyms of the pieces, all of them have written the figure number of *La antigüedad* . . . in white (Figure 2). This characteristic was recorded on the photograph of the publication by Lehmann-Nitsche, which allows us to think that those references were there since that moment or before,

because, as we pointed out, pieces with this type of acronym had also been recorded in the Archeology Division (Figure 3).

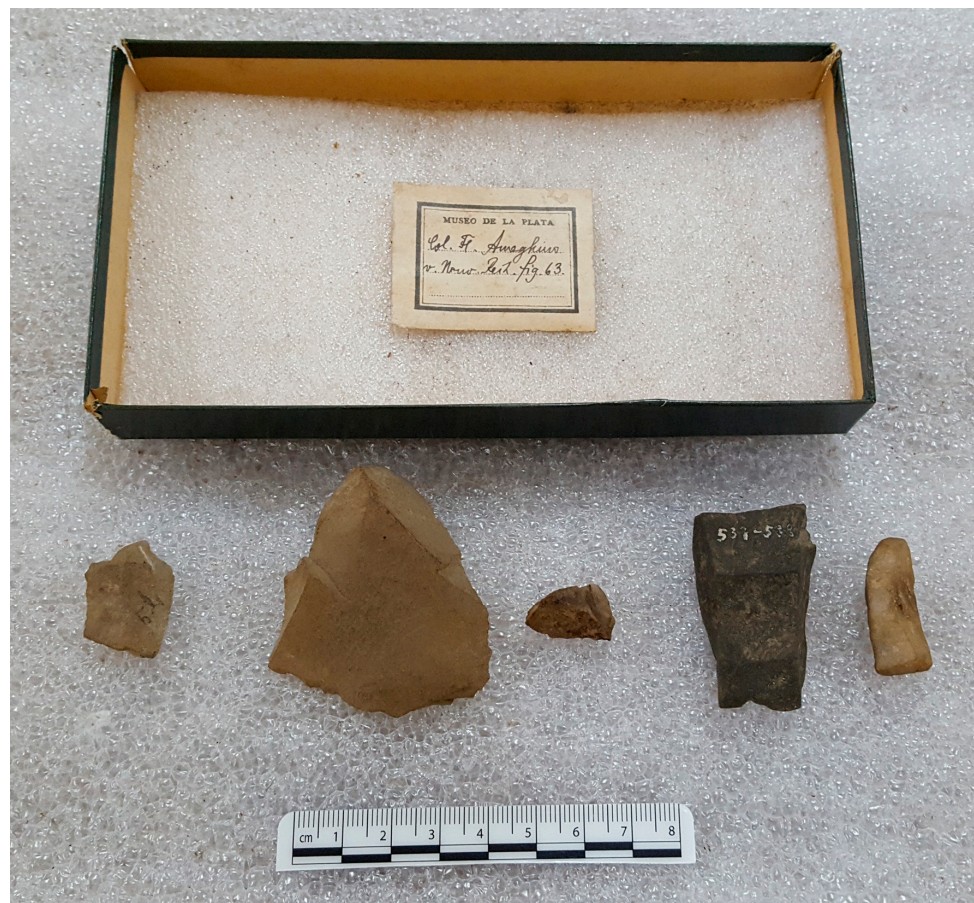

**Figure 2.** Set of lithic artifacts stored in the Anthropology Division of the Museum of the La Plata Museum labeled with the letterhead of the museum and the following inscription: "Col. Fl. Ameghino v. Nouv. Rech. Fig. 63".

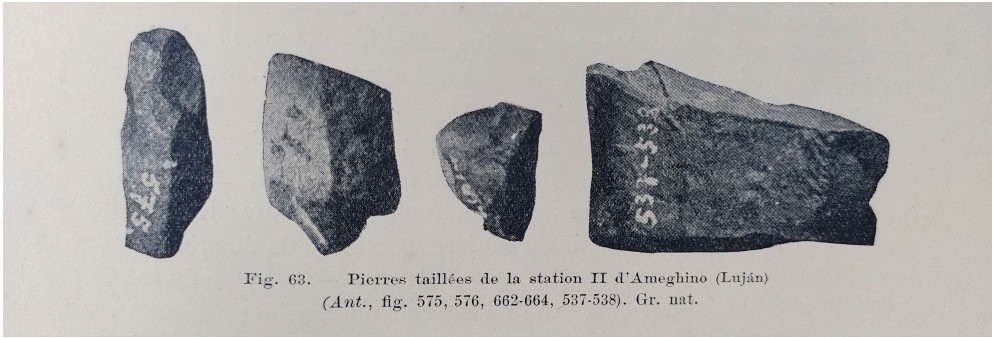

**Figure 3.** Set of lithic artifacts stored in the Anthropology Division (see Figure 2), that were presented as photographs in Lehmann-Nitsche's paper for the La Plata Museum Journal in 1907.

The third identified criterion corresponds to the *Catálogo de las colecciones arqueológicas y etnográficas* {Archeological and Ethnographic Collections Catalog}, information for the museum's internal use, written by Luis María Torres (1878–1937) in 1915 when he was in charge of the Archeology Section of the La Plata Museum [42]. It involves numbers that range from 12,445 to 13,915, written in black ink or pencil on the lithic artefacts of plates from I to XVI. In Figure 1, the number 13,063 is related to this catalog. Based on the analysis of the site's numbering and descriptions provided by Ameghino on *La antigüedad* . . . , we

could determine some mistakes regarding the proveniences that later Torres recognized, since he established Chascomús and Lobos as the origin of objects that in fact came from Cañada de Rocha.

Once the pieces were identified, we established a new arrangement criterion according to the plates of *La antigüedad . . .* , making sets of objects in agreement with the layout of the illustrations on the plates. All of those materials were prepared by Julieta Pellizzari (conservator of the Archeology Division) on polyethylene foam sheets, which were cut according to the morphology of the objects and lined with tyvek fabric. All of them were placed in new containers, acid-free, and labeled with the numbers that match the identified plates and figures at the same time that we preserved all the references found at the time of the identification. Carolina Silva (archeologist of the Archeology Division) produced the systematic photographic recording of the Ameghino Collection for the digitalization of each piece and their inclusion into an open access digital repository. All of this was carried out for the purpose of making the information public, reducing the handling of the collection, and facilitating its remote consultation for research (Figure 4).

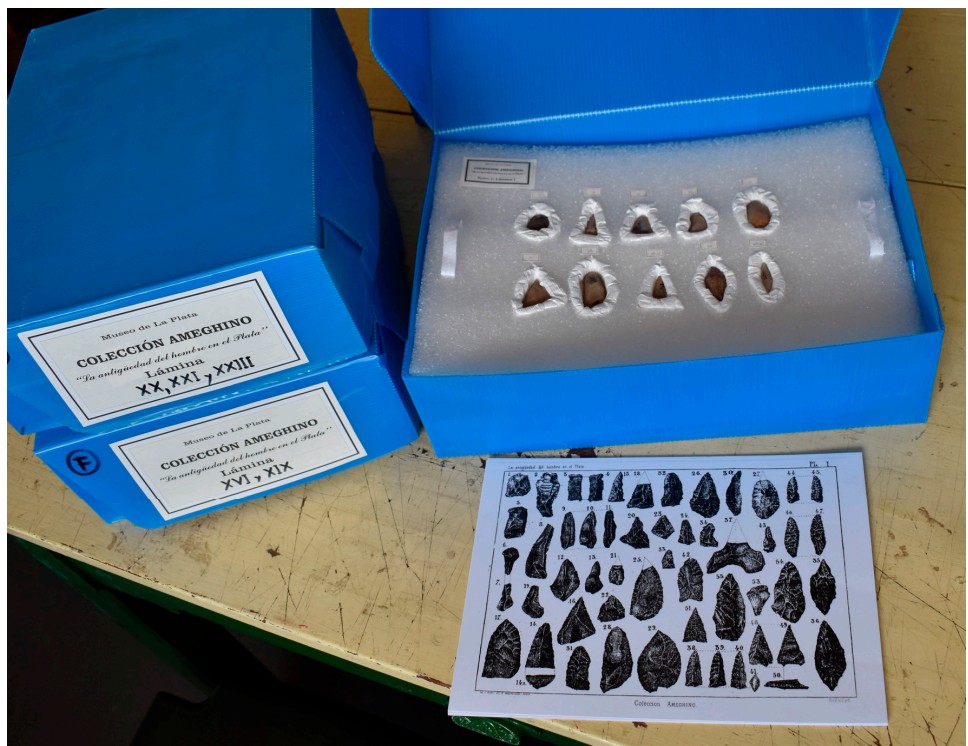

**Figure 4.** Pieces in their new containers, arrange according to the plates of *La antigüedad del hombre en el Plata*.

## 3. Ameghino's Collection Objects: Proveniences and Artifact Analysis

Regarding the study of the objects, once they were identified, we analyzed them, distinguishing sets by material type. Thus, 49% of the represented percentages corresponded to bone materials (n = 93), 28%, to lithic artifacts (n = 53), and the remaining 24%, to pottery (n = 45). Based on this first classification, we carried out the formal analysis of the materials, as well as the proveniences identification.

### 3.1. Lithic Artifacts

The techno-morphological study of the lithic materials was conducted in accordance with the guidelines by Aschero and Bellelli et al. [43,44]. The lithic artifacts are made up of tools (n = 36), debitage (n = 15), and cores (n = 2), produced in chert (n = 24) and orthoquartzite of the Sierras Bayas Group (n = 21) [45], and to a lesser extent, of other raw materials (n = 8) such as silex, basalt, and xylopal. Sizes medium small (n = 22) and small

(n = 13) are predominant, while the predominant length-wide relation are those medium normal (n = 13) and medium long (n = 13), followed by the laminar normal (n = 9). Debitage is an assemblage of fractured (n = 9) and complete (n = 5) lithic flakes of an angular, crest flat, and bipolar types with plain, filiform, dihedral, point-like butts, and mainly diffuse and undifferentiated bulbs. We also recorded an unidentified lithic flake and two cores, one over a nodule.

The tools are made up mostly of retouched flakes with sharp asymmetric edges (n = 12), which are bilateral, frontolateral, and lateral, and also of side-scrapers (n = 8) with double convergent, bilateral, and lateral cutting edges, as well as scrapers (n = 7) of frontolateral, perimetral, and lateral edges. The remaining are knives, perforators, small triangular unstemmed projectile points, a bifacial artifact, unidentified fragments, and a pendant. The tools were made mainly through unifacial (n = 28), short marginal (n = 25), or ultra-marginal (n = 19) retouches and micro-retouches on angular lithic flakes (n = 17). It is in informal tools (n = 27) [46] that the geometrical shapes of the edges are variable, among which triangular (n = 6) and rhomboidal (n = 5) shapes stand out. In two instances, we observed a lateral sharp edge and a notch on composite tools.

### 3.2. Pottery

Pottery is essentially made up of body (n = 17), and rim fragments (n = 15), part of vessels with rim, body and/or bottom (n = 3), solid strap handles (n = 5), fragments of "tubular" pottery (n = 3) [47], aspindle whorl, and the chamber of a smoking pipe. In two fragments, we identified a spout and holes for hanging/repairing. The majority of the ceramic is decorated with line-shaped incisions (n = 22), mainly parallel or crossed forming rhombus. We also recorded stippling, fingernail and fingertip impressions, and red paint.

### 3.3. Bone Remains

Out of all the bone materials identified (n = 93), at an Order level, we managed to recognize four groups: Artiodactyla (n = 17, mainly Cervidae), Carnivora (n = 1, mainly Felidae), Notoungulata (n = 10, for Toxodon) and Pilosa (n = 1, for Mylodon). We identified some parts of the skeleton that included fragments of skull, teeth, antlers (n = 3), and postcrania which included long bones, among which there was a tibia and a femur, ribs, a vertebra, a scapula, and metapodials, as well as other parts that we could not identify due to their fragmentation.

Regarding the modifications of anthropic origin, we could identify them in a significant number of pieces: cut marks and thermal alterations, as well as many specimens with fresh fractures, mainly of the spiral type, many of which have impact points and/or negative flake scars. Fifteen of these bone artifacts had tiped or blunt apices, in some cases these showed clear manufacture evidence (polish, debitage, and roughing) which make fluted and flat points (n = 5) on long bones of mammals. Moreover, we recorded a Cervidae antler fork with polished edges, and was hollow due to the elimination of spongy tissue.

### 3.4. Provenience of the Materials

The reconstruction of the provenience of the artifacts and bones identified in the collection was carried out based on the analysis of the details provided by Ameghino in *La antigüedad . . .* , as well as the data included in volumes II and XIX of the *Obras y correspondencia científica* {Scientific works and correspondence}, edited by Alfredo Torricelli [48,49]. Ameghino used the period classification that characterizes the 19th century's conception of European Prehistory and used the term "paradero" {stop point}, which was a frequently used category in the Argentinean literature of that period, picking up a concept that was originally presented by another Italian natural scientist, Pellegrino Strobel (1821–1895), to name the places or stations that indigenous populations seasonally frequented for specific activities in times after or immediately before the European colonization, such as residential camps or stone quarries, among others [50].

The amount of detail on the provenience of each of the objects provided in the descriptions in Ameghino's *La antigüedad . . .* was quite varied, due to the focus of the wording being placed on the technical description of the materials. Many of these objects were collected by Ameghino, but others had been handed to him by collectors and contacts. Specifically, he claimed that, "we have gathered remains of the ancient Indian history in the North, South, and West of the [Buenos Aires] province, but the only paraderos that we have carefully explored and in which we have performed regular excavations are located near Mercedes and Luján, areas where our usual residence simplified to our work" [1] (p. 302). However, a great percentage of the objects lacked any details of their origin, and he stated that "the main places in which they have been found are: Buenos Aires and its surroundings, San José de Flores, Villa de Luján, Pilar, San Antonio de Areco, Salto, Ensenada and almost the entirety of the Atlantic coast, the mouth of the Salado River, in Puente Chico, near Barracas, Chascomús, Tandil, and finally even in the Laguna del Monte . . . " [1] (pp. 215–216).

Within the relative chronology proposed by Ameghino, most pieces identified in the collection corresponded to the "Paleolithic" (39%; n = 75) and the "Neolithic" periods (36%; n = 70), and, to a lesser extent, to the "Mesolithic" period (24%; n = 46) (Table 1). All the materials came from the Buenos Aires Province, and within this group, most of them (n= 10) were located in the middle basin of the Luján river, in the districts of Mercedes and Luján (Figure 5). Among these proveniences, 46 pieces corresponded to the Cañada de Rocha archaeological site, while 74 origins corresponded to paraderos 1, 2, 3, 4, 5, 6, and 7 of Lujan River (n = 74), especially number 2 (n = 30) and number 4 (n = 17). We also recorded pieces found in the Frías creek (n = 4); Balta creek (n = 1); Areco river (n = 1); the area of the Villa de Luján (n = 1); in the "Barrancos field" located at two leagues from the Mercedes (n = 1); and in the archeological site known as the Bretón brother's (n = 1), since it had been studied by a couple of French collecting enthusiasts of that region.

**Table 1.** Materials distribution according to Ameghino's relative chronology. They are organized by material type, proveniences and plates.

| Ameghino's Relative Chronology | Provenience | Plates of *La antigüedad . . .* | Lithic | Pottery | Bone Remains | Total |
|---|---|---|---|---|---|---|
| | Paradero de Olivera (Lujàn) | I | 1 | 0 | 0 | 1 |
| | San Antonio de Areco | I | 1 | 0 | 0 | 1 |
| "Neolithic" | Prov. de Bs. As. (Luján, Mercedes, Ciudad de Buenos Aires, San José de Flores, Pilar, San Antonio de Areco, Salto, Ensenada y casi toda la costa del Atlántico, la embocadura del Salado, en el Puente Chico, cerca de Barracas, Chascomús, Tandil, Laguna de Monte) | I, II, III, IV, V, VI, VII, VIII | 23 | 37 | 0 | 60 |
| | Paradero de Luján (Luján) | II, VI | 1 | 1 | 0 | 2 |
| | Paradero del Arroyo Frías (Luján) | II, VI | 1 | 1 | 0 | 2 |
| | Paradero del Arroyo Frías (Mercedes) | II, IV, VII | 4 | 1 | 0 | 5 |
| | Campos de Barrancos (Mercedes) | II | 1 | 0 | 0 | 1 |
| "Mesolithic" | Paradero de Cañada de Rocha (Luján) | XIII, XIV, XV, XVI | 8 | 6 | 32 | 46 |

**Table 1.** *Cont.*

| Ameghino's Relative Chronology | Provenience | Plates of *La antigüedad* ... | Lithic | Pottery | Bone Remains | Total |
|---|---|---|---|---|---|---|
| "Paleolithic" | Paradero 1 (Mercedes) | XIX, XX, XXIII | 4 | 0 | 3 | 7 |
| | Paradero 2 (Luján) | XIX, XX, XXI, XXII, XXIII, XXIV, XV | 4 | 0 | 27 | 31 |
| | Paradero 3 (Mercedes) | XIX, XX, XIV | 3 | 0 | 4 | 7 |
| | Paradero 4 (Mercedes) | XIX | 0 | 0 | 17 | 17 |
| | Yacimiento Hnos. Bretón (Luján) | XIX | 1 | 0 | 0 | 1 |
| | Paradero 5 (Luján) | XX, XXI, XXII, XXIII | 0 | 0 | 9 | 9 |
| | Paradero 6 (Mercedes) | XXIII | 2 | 0 | 0 | 2 |
| | Paradero 7 (Mercedes) | XXIV | 0 | 0 | 1 | 1 |
| **Total** | | | 53 | 45 | 93 | **191** |

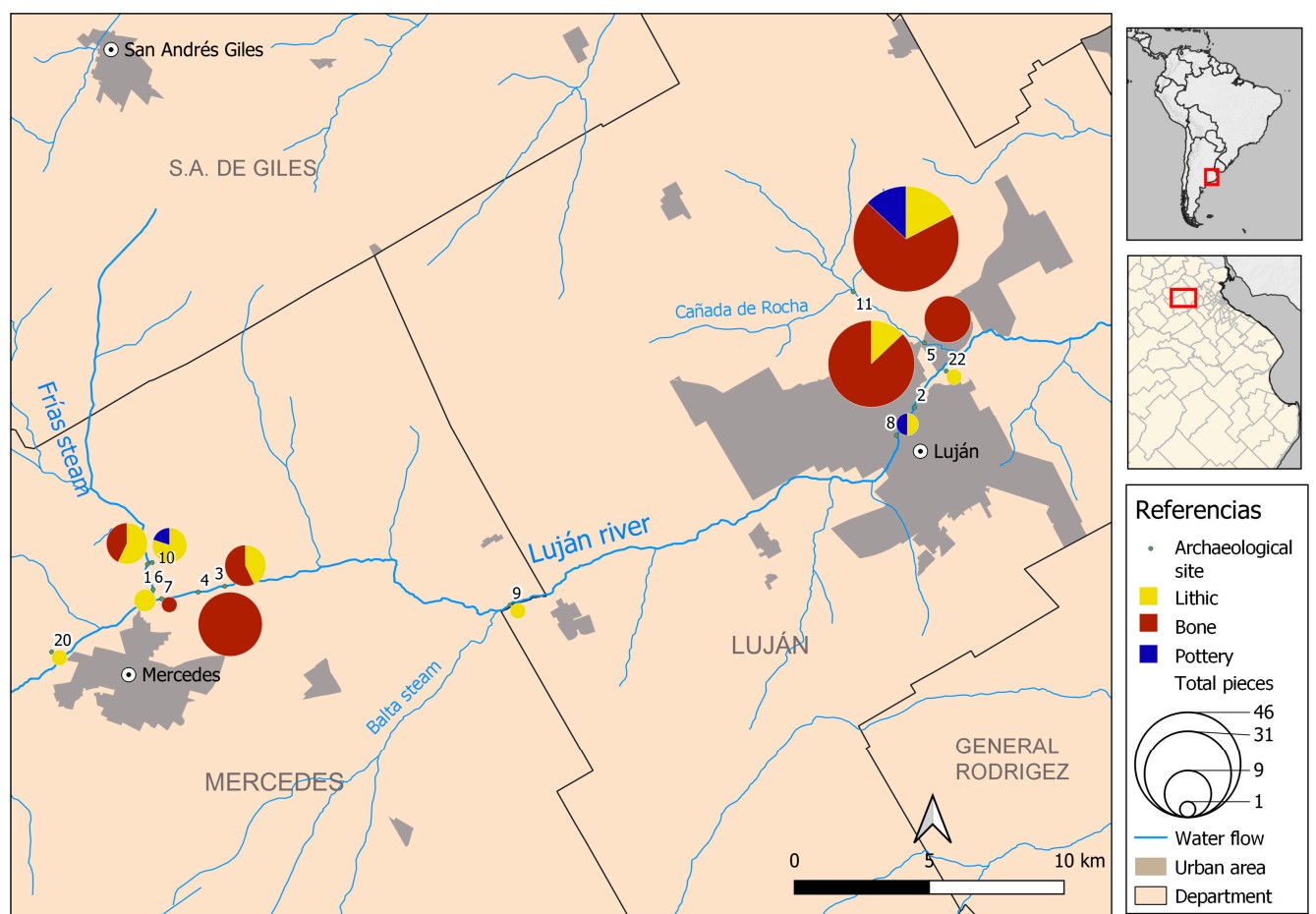

**Figure 5.** Archaeological finds placed at the basin of Luján's river (Buenos Aires province, Argentina). The pieces are presented by material type (yellow: lithic, red: bone, and blue pottery) and quantity. References: 1—Paradero 1; 2—Paradero 2; 3—Paradero 3; 4—Paradero 4; 5—Paradero 5; 6—Paradero 6; 7—Paradero 7; 8—Paradero de Luján; 9—Paradero de Olivera; 10—Paradero del Arroyo Frías; 11—Cañada de Rocha; 20—Campos de Barrancos (Mercedes); 22—Yacimientos Hnos. Bretón (Luján).

### 4. Comparative Analysis of the Objects of the Ameghino Collection and the Illustrations from *La Antigüedad* . . .

The illustrations originally published in *La antigüedad* . . . were arranged in plates imprinted on stone matrices at the end of each volume. They were preceded by a list of figures with a label with the image number, the sheet number, and the page number corresponding to the text where the explanation was provided. This contributed to the embedding of the image in the text, but also indicated a possible, strictly visual reading, allowing the images to become an independent entity. This idea is strengthened by other formal data written in the plates; all of them were numbered and most of them read "AMEGHINO Collection", which indicated to whom the objects belonged. The name of the designer, Zacarías Bommert, the name of the book, and, in some cases, the information of the printing house were also included (Figure 6). Regarding the comparative analysis of the collection with these illustrations, those objects identified were distributed in 18 plates out of 25. These plates include the representation of one through 52 objects in total, with an identification range per plate of 100 to 12%. The seven remaining sheets have illustrations of stratigraphic sections and materials coming from other collections that Ameghino consulted for his book, but are not part of his collections.

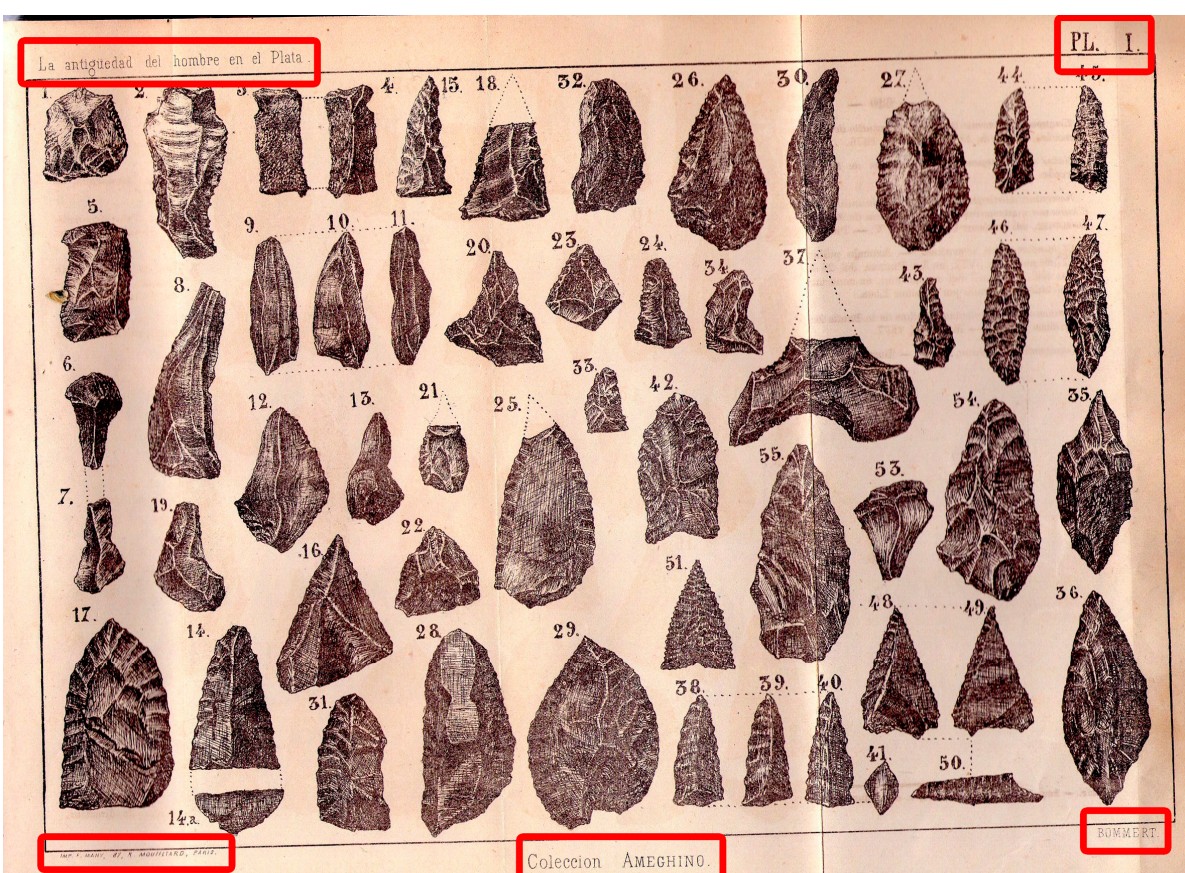

**Figure 6.** Plate I from *La antigüedad* . . . Marked in red, an example of all the indications presented in the plate of figures from the book of Ameghino . . . Above, the name of the book and the number of the plate. Below, the name of the printer, the owner of the collections and the designer of the plate.

The arrangement of the book contents followed temporal and spatial criteria that complemented each other and articulated in favor of the main thesis: demonstrating that people had coexisted with extinct megafauna in the Pampas. The argumentative structure followed the chronological sequence of the human occupation of the Argentine territory, from least to most antique, using the categories and the prevailing terms in the studies from European Prehistory of that time (see Table 1). Regarding the division of the "archeological

eras," in the first place, he presented the information on the most modern archeological sites, "Neolithic" period, which would correspond to the populations that had developed since the beginning of the modern geological period, the "modern alluvium" in Ameghino's terminology. They had well-developed stone and pottery industries and had been contacted by the European colonizers in the 16th century. After that came the information about the "Mesolithic" paraderos, which according to Ameghino were more antique, shown in their geological context, "Late Quaternary," and in the development of pottery and lithic manufacturing technology that was rougher than the previous one. Finally, the "Paleolithic" and "Eolithic" paraderos were jointly represented by cultural materials and specimens of extinct megafauna, mainly Mylodon, Toxodon, and Glyptodon, and were geologically situated in the "Early Quaternary" and the "Late Tertiary" periods, respectively.

The images of the book were adjusted to that time criteria, taking place according to the storyline to support the arguments, but also to prepare the reader to recognize each feature to be able to understand the artifacts and modified bones as a result of the action of an intelligent being that had inhabited the Pampean region in a remote past. For that purpose, Ameghino used varied resources, such as explaining the properties inherent to the objects; for example, the characteristics of the rocks or the placement of the cutting edges in the case of lithic objects, or the types of marks, shapes, sizes, and gestures, in the case of megafauna bone remains. In other cases, he also resorted to the morphological similarities that his materials had with those described in the archeological literature of European Prehistory, or he specified the location within old strata.

The represented materials follow the general structure of the work, first describing and illustrating the lithic artifacts from the Buenos Aires Province. Then, those that came from the indigenous occupations immediately before contact with the European colonizers, and finally, the remains of technology linked to fossil fauna. In some plates, the objects belonged to different but consecutive chapters. These decisions bring continuity and fluency to the argument, while they reduce the design and image publication costs, making the most of the space on the plates to illustrate as many objects as possible [5].

Each object is drawn in its original size and, in most cases, illustrated by only one of its faces, the one that showed the most evidence of human intervention, such as the negative flake scars in the lithic artifacts or the decorative incisions on the pottery. By doing it this way, the representation of a broad set of collection pieces was achieved, keeping the aforementioned book-catalog format. Choosing several illustrations for a single piece had a pedagogic criterion: train the sight of the reader though the explanation and exemplification of the prominent characteristics of the represented materials that showed human intervention, such as cutting, scraping, or drilling tools, the bones with cut marks or spiral fractures in the diaphysis of long bones.

However, based on the formality of the illustration, those qualities in the materials that the reader should recognize were considered, as the figures were studied according to the formal elements of the representation conventions to achieve realistic images of each of the pieces [51]. The use of hatching on the drawing to show saturation, fills, and outlines produced light or shadow effects, simulating the typical textures of the represented materials: lithic, bone, or ceramic. Those resources allowed us to highlight the content of the drawing, especially those epistemic qualities, which are necessary to strengthen the argument, favoring the material recognition apart from the text, based on its own aesthetic and technical representation values [52].

When comparing these illustrations to the originals, these ideas became enhanced. In the case of lithic artifacts, out of the assemblage, the ones particularly illustrated were those of a laminar shape, the edges of which were retouched and whose illustrations were oversized, as in the case of a side-scrapers represented in figure 80 of plate number II (Figure 7). When we observed the object displayed in the same direction as in the figure, the overall shape, the outline, its transparent coloration and a few negative flake scars, and a short, sharp edge were highlighted. Conversely, in the drawing, the retouches of the edge are a little accentuated. This detail is emphasized by the description of this unifacial

piece, in which that feature is especially stated, since "its inferior side is completely flat and concave. The superior side is convex and also flat, but its left edge is about five millimeters wide, and it was carved by spiral percussion so that a sharp edge is produced" [1] (p. 234).

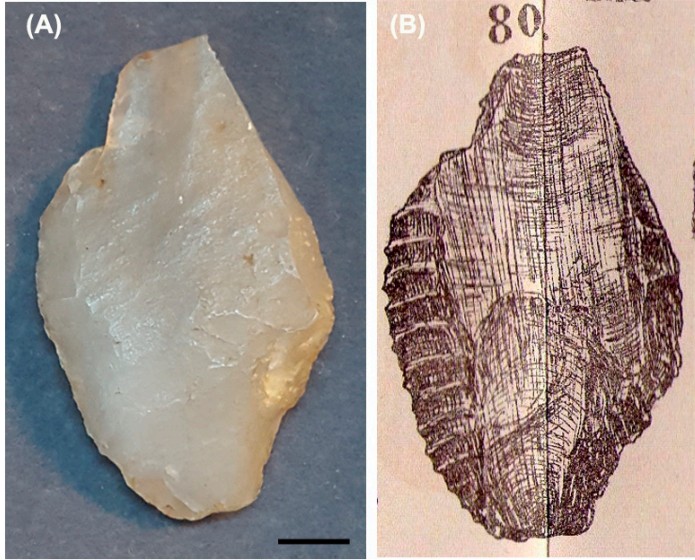

**Figure 7.** Lithic tool. An example of the use of the formal elements of the representation conventions to give a realistic image to lithics materials. An original flake with a sharp edge (**A**) and its illustration, published in *La antigüedad* . . . , in which the lateral cutting edge is oversized (**B**). Scale bar: 1.5 cm.

For the pottery, due to most of it being sherds, the edges and outlines of those decorated with incisions were illustrated with a lot of precision. The information in the text and the images follows the description of the most important aspects of the sherds: thickness, coloration, paste firing, and decorative designs. The latter were classified by shape and ornamentation. According to the comparison between the drawings and the materials, we highlight the relevance given to the details, depicting the deep incisions on the edge through a closely spaced hatching and fine lines that indicate the irregular outline (Figure 8).

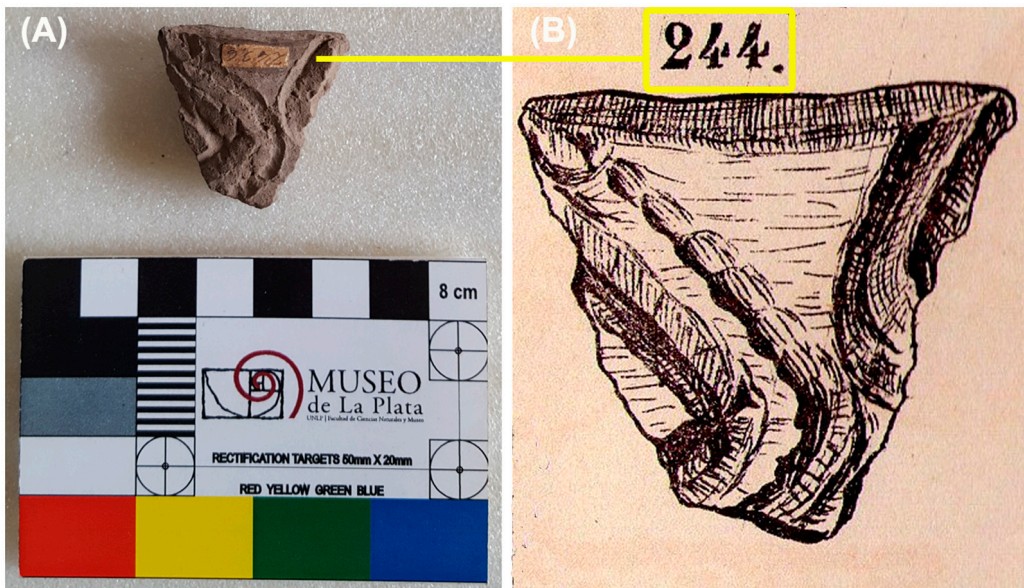

**Figure 8.** Pottery sherd. Another example of the use of the formal elements of the representation conventions to give a realistic image to pottery. In this case the importance of the details is highlighted by comparing the illustration (**B**) with the original piece (**A**).

It is in the case of the figures of bones where the didactic function of the illustrations and their persuasive aspect is the most evident, especially to highlight human action. The depictions of broken or burnt bones, as well as those with tiped, carved ends which were used as instruments, especially points, served as a means to provide meaning to the argument. For such a purpose, and as seen in the previous examples, pictorial resources using hatching highlighted the volume and shape of the material, its coloration, and the suggested cut marks, points of impact, and flake negatives. The arrangement of the figure within the illustrated materials on the plate worked in the same way. The progressive placement of the objects as figures, starting with those with well-defined shapes such as stone instruments, were explained in the text to support visual training. This way, when reaching the most undefined pieces, the reader already had some training in detail recognition.

This is especially noticeable in the record of impact and cut marks on bones of extinct fauna. In *La antigüedad* . . . , one of the most important pieces of evidence of the coexistence was the illustration of a Mylodon tibia, on which Ameghino had previously sought experts' opinions and received the aforementioned positive verdict. In the text, he presented a very detailed description, explaining the differences between the marks, the ways in which they had been produced, and their shape characteristics, pointing out the differences between furrows and incisions. In the three figures, made in natural size and which represented the diaphysis and epiphysis of the tibia, he established a set of letters to identify every modification observed in the bone (Figure 9).

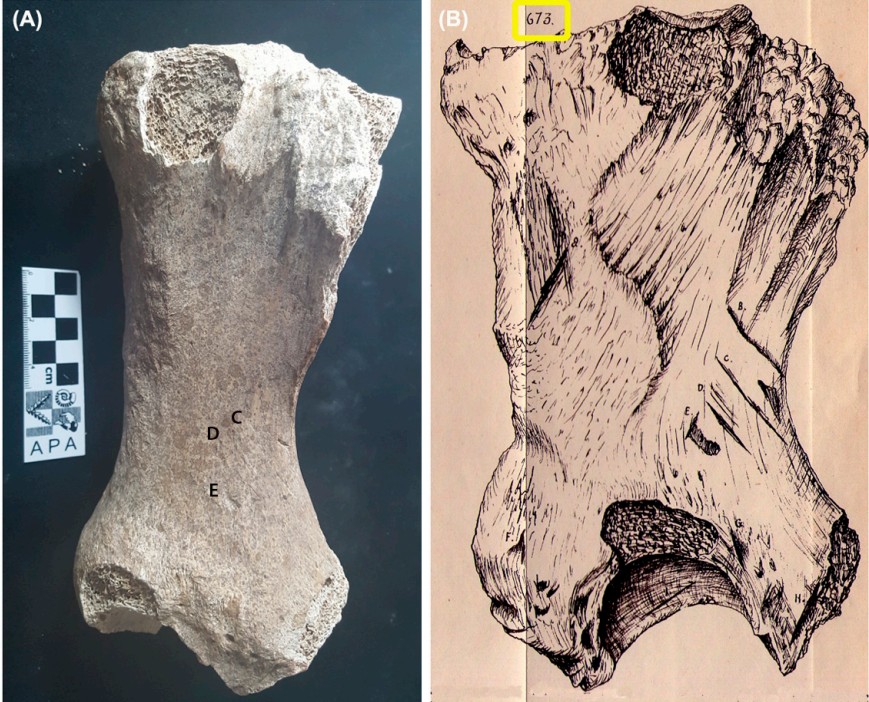

**Figure 9.** Mylodon tibia that was used for Ameghino as the main evidence of the coexistence of humans with fossil fauna (**A**). The capital letters C, D and E represented the marks produced by people described by Ameghino in *La antigüedad* . . . The illustration of this tibia was originally published in *La antigüedad* . . . (**B**), with the aforementioned marks.

## 5. Conclusions

Visual content was a fundamental aspect in *La antigüedad* . . . , as it provided the structure for all the information and the author's arguments. Their editing and distribution made the collections known and trained the reader's view to recognize and identify prehistoric materials. The development of these discursive and visual strategies allows us to include *La antigüedad* . . . in the history of book-catalogs, characterized as illustrated mobile

museums in which pieces from the natural or cultural world were displayed in order and classified according to previously established scientific criteria [53]. Created and published for different purposes, these books used to have high quality editions, presented many images, and served as a reference material database or empiric corpus for research. At the same time, they were presented as catalogs for the sale of collections and were a means to spread scientific ideas, as they showed examples and, especially the images themselves, were empirical evidence of the issues brought forward. In this sense, *La antigüedad ...* is closely linked to the main publications from the 19th century European Prehistory, such as *Pre-historic Times by ancient remains and the manners and customs of modern savages* (1865) by John Lubbock or the publications by Gabriel de Mortillet, especially the commented catalog of the Saint Germain Museum collections, *Musee Prehistorique* (1881), which he published with his son Adrien [54,55]. The similarities are observed in their choice of a temporal sequence, by ages and periods, according to chronostratigraphic stages with fauna proxies as fossil guides. Furthermore, the editing and processing of the images are examples that confirm the variety of mechanisms that were used in the presentation and use of the scientific illustrations in the 19th century. These conclusions greatly promoted the search, identification, and rearrangement of the Ameghino Collection of the La Plata Museum, weighing in the 21st century the historical value of the articulation between the objects and the book where they were published for the institution. Those same assessments, within today's context of the museum where they are kept, as well as the state of the studies in science history, have enabled us to think and suggest the actions that favored its value recognition, conservation, and diffusion in a digital repository.

The works carried out on the initial collections that Ameghino assembled during the 1870s allow for the confirmation that those materials have been kept in the La Plata Museum for over 130 years once they had returned from Paris, where they had been part of several scientific events. These are the historical objects with which the scholar put forward the coexistence of human beings with Pleistocene megafauna and that were published as figures in his main archeological publication, *La antigüedad ....* Considering the visual content and the information present in the work, we recognized the high integrity of the collection, highlighting that many of the pieces are in a good general preservation state. Based on the artifact analysis, we emphasize the tendency to represent the decorated sherds with incisions, the lithic retouched artifacts, and the diaphysis of long bones with laminar shape and points. Regarding the proveniences, materials were collected from various sites near the basin of the Luján River, allowing for the furtherance of the investigations in this area, about which there has been little discussion during the 20th and 21st centuries.

The development of this interdisciplinary investigation has enabled us to revisit the objects and Ameghino's work with new perspectives for the investigation of the history of the collections and the archeological practices. The purpose of these experiences is to consider the historical value of Florentino Ameghino's initial collections, for the museum and historic preservation practices of the La Plata Museum, as well as for studies in science history. In this sense, we promote the historical value of this book-catalog due to its quality and the detailed record of the materials. Finally, the furtherance of this line of research brings us the possibility to move forward with new forms of public communication of science and to develop interfaces between art and science for heritage activation.

**Author Contributions:** Conceptualization, C.S., M.B., S.L. and G.A.S.; methodology, C.S., M.B., S.L. and G.A.S.; formal analysis, C.S., M.B., S.L. and G.A.S.; investigation, C.S., M.B., S.L. and G.A.S.; resources, C.S., M.B., S.L. and G.A.S.; data curation, C.S. and M.B.; writing—original draft preparation, C.S., M.B., S.L. and G.A.S.; writing—review and editing, C.S., M.B. and S.L.; visualization, C.S., M.B. and S.L.; supervision, C.S., M.B. and S.L.; project administration, C.S.; funding acquisition, M.B. and S.L. All authors have read and agreed to the published version of the manuscript.

**Funding:** This research was conducted thanks to the following projects: PUE 2017-0002 "Colaboratorio por la diversidad cultural. Integración multidisciplinaria para el desarrollo y resolución de problemáticas sobre materialidad, patrimonio y pluriculturalidad" (IDECU-UBA-CONICET), "Arqueología de ambientes acuáticos del Centro-este argentino" (UNLP-11/N885) and "La práctica y

la comunicación de la arqueología bonaerense en el siglo XX y XXI: espacios institucionales, saberes académicos y comunidades locales del sur de la provincia de Buenos Aires" (SECYT-UNS, 24/I276).

**Data Availability Statement:** The data presented in this study are available on request from the corresponding author.

**Acknowledgments:** Project D395/21 "Digitalización de las colecciones de la División Arqueología del Museo de La Plata (Facultad de Ciencias Naturales y Museo, Universidad Nacional de La Plata)", funded by Conicet, Fundación Bunge y Born, and Fundación Williams. Project PIP 11220200101844CO "Dinámica ambiental y uso del espacio en la cuenca del río Luján desde la transición Pleistoceno-Holoceno hasta el siglo XVI" funded by Conicet and Project PUE 2017-0002 "Colaboratorio por la diversidad cultural. Integración multidisciplinaria para el desarrollo y resolución de problemáticas sobre materialidad, patrimonio y pluriculturalidad" (IDECU-UBA-CONICET).

**Conflicts of Interest:** The authors declare no conflict of interest.

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
