# Peer review of "Materiality and Images: Ameghino’s Collection of La Antigüedad del Hombre en el Plata in the La Plata Museum"

_heritage, doi:10.3390/heritage6020086_

Round 1

Reviewer 1 Report

In general terms this study is interesting and deals with one of the most important collections in Argentina, as well as the reference book La Antigüedad in which this collection was mainly published. 

In lines 31 and 61 the La Antigüedad is cited but from different editions. It would be convenient to only use one or otherwise mention why two are used. On the other hand, in this kind of study it is convenient to use the original works and not subsequent editions. Unless it was impossible for the author to access to the original edition.

[1.- Ameghino, F. The antiquity of man in Plata, 1st ed.; IGON Brothers: Buenos Aires, Argentina, 1880-1881

5.- Ameghino, F. The antiquity of man in La Plata, 1st. Ed.; Intermundo: Buenos Aires, Argentina, 1947, volume I.]

------------------------------------------------------------------------------------

The example of catalog numbers shown in figure 1 is not clear. It should be indicated in the image to which number each case refers. Four numbers appear on the bone, 452 twice, 660 and 13063. All of them should be mentioned in the text and referenced in the figure (for example: I understand that 13063 would correspond to the Torres numbering). It would also be desirable if the images only contained the elements that are being studied such as bones and that they were in the same position or view, if this is possible. Figure 1 should show that the bone in image A is the same bone as that in images B and C but from another view. All figures must be to scale.

------------------------------------------------------------------------------------

Lines 82 to 90 require bibliographic citations. In particular, those mentions referring to the sale of the Ameghino collection to the Museo de La Plata and his incorporation in Buenos Aires after his return from France, in which this collection had a remarkable role.

(Podgorny, I. 2009. El sendero del tiempo y de las causas accidentales. Los espacios de la prehistoria en la Argentina, 1850-1910. Prohistoria ediciones, Rosario, 331 pp.

Podgorny, I y Lopes, M.M. 2008. El desierto en una vitrina. Museos e historia natural en la Argentina, 1810-1890. Editorial LIMUSA, Mexico D.F., 279 pp.

Podgorny, I. 2011. Los Reyes del Diluvium. La geología del Cenozoico sudamericano en la década de 1880. Asociación Paleontológica Argentina, Publicación Especial 12: 21-34.

Farro, M.E. 2009. La formación del Museo de La Plata. Coleccionistas, comerciantes y naturalistas viajeros a fines del Siglo XIX. Prohistoria ediciones, Rosario, Argentina, 230 pp.)

------------------------------------------------------------------------------------

Figure 8. The number 244 assigned to the piece drawn in the book La Antiguedad seems different from the number of the specimen in the archaeological collection of the Museo de La Plata

------------------------------------------------------------------------------------

Figure 9: The view of the bone in the photograph of figure 9 should correspond exactly to the drawing of the same bone. In addition, the marks mentioned in the text could be indicated.

------------------------------------------------------------------------------------

lines 456 to 458 "This is especially noticeable in the record of impact and cut marks on bones of extinct fauna. In Antiguedad…, the main evidence of the coexistence was the illustration of a Mylodon tibia, on which Ameghino had previously sought experts' opinions and received the aforementioned positive verdict."

It could certainly be true that some evidence is more important that other, but this claim should be further discussed as Ameghino's study analyzes evidence (Stratigraphic, archaeological, paleontological, etc.) as a whole and not individually.

------------------------------------------------------------------------------------

Lines 505 a 508 Such was the impact of this book on the country's archeology, that the Asociación de Arqueólogos Profesionales de la República Argentina {Argentine Association of Professional Archeologists} established Florentino Ameghino's birth date (September 18th) for the celebration of the day of the Argentine archeologist.

            The authors could evaluate if these lines should be in the introduction rather than in the conclusions.

------------------------------------------------------------------------------------

Author Response

We are deeply thankful for the revision of our manuscrip. It surely helped us to improve our work. We have accepted and corrected all your comments. You can see them in the attachment.

Regards,

Reviewer 2 Report

Overall, this is a great example of the reexamination of archaeological archival materials through multiple lines of evidence. The English is very good for the most part, with some editing necessary in pluralisation in particular, for example lines 49-50, line 55, etc. Excellent use of visual arguments in, for example, Figure 1, well done. The techniques used in illustration of the artefacts in La antigüedad are non-standard and quite interesting with regard to visual lineages of illustration in archaeology. I would be interested in a discussion comparing these conventions to other traditions, such as those in Archaeological illustration by Adkins and Adkins, but it is not necessary, as this article (with some English editing) can be published as-is. Thank you for bringing this work in archaeological historiography and visual argumentation to an international audience.

Author Response

(The authors gave the same response as above.)

Reviewer 3 Report

Line 47. For consistency, I would add dates to Ramorino: (1841-1876)

Line 55. Specify where the Exposition took place.

Line 72. Please add reference to classification.

Line 206. For consistency, I would add dates to Torres.

Line 201-202 (for Editors). I would expect to have the caption in the same page of the relevant image.

The paper is clear in the exposition and provides context on Florentino Ameghino, his collection and works, in particular the book “La antigüedad del hombre en el Plata”, important reference as a 19th century History of Science book for its description and images of the artifacts but also to the perspective of scientific information dissemination (i.e. the use of drawings for specific purposes). It is supported with a strong reference list and with its conclusions it adds advanced research on the Amerigo’s collection in the Museu de L Plata. I particularly found the comparative analysis useful for the purpose of the paper and to clarify the different inventory numbers. The analysis of the objects and added information about their place of origin is plainly described. The only consideration I would have is the use of the term “provenance” as “provenience”, when meaning the specific location where an object was unearthed. The paper provides details about the history of the collection, the chronology of the ownership, from its assembling to their current location (i.e. provenance; e.g. on page 14), so I would use the term “provenience” for referring to the objects’ findspot. I understand this is a current discussion in the cultural heritage field, with some confusion around them and the terms are used also interchangeably, but because we are considering objects in the La Plata Museum, I would prefer to see “provenience” instead.

Author Response

(The authors gave the same response as above.)
